# Analysis of the Effect of Mixed Fermentation on the Quality of Distilled Jujube Liquor by Gas Chromatography-Ion Mobility Spectrometry and Flavor Sensory Description

**DOI:** 10.3390/foods12040795

**Published:** 2023-02-13

**Authors:** Busheng Zhang, Zhongguan Sun, Liangcai Lin, Cuiying Zhang, Chunhui Wei

**Affiliations:** 1Shandong Engineering Technology Research Center of Pomegranate Deep Processing, College of Food Science and Pharmaceutical Engineering, Zaozhuang University, Zaozhuang 277100, China; 2Key Laboratory of Industrial Fermentation Microbiology, Ministry of Education, College of Biotechnology, Tianjin University of Science and Technology, Tianjin 300457, China; 3Liquor Making Biological Technology and Application of Key Laboratory of Sichuan Province, Zigong 643002, China

**Keywords:** jujube liquor, volatile, GC-IMS, PLS-DA, E-nose

## Abstract

Distilled jujube liquor is an alcoholic beverage made from jujube, which has a unique flavor and a sweet taste. The purpose of this study was to explore the effect of mixed fermentation on the quality of distilled jujube liquor by comparing the performance of mixed fermentation between *S. cerevisiae*, *Pichia pastoris* and *Lactobacillus*. The results showed that there were significant differences in the quality of the jujube liquor between the combined strains. Moreover, *Lactobacillus* increased and *P. pastoris* reduced the total acid content. The results from an E-nose showed that the contents of methyl, alcohol, aldehyde, and ketone substances in the test bottle decreased significantly after decanting, while the contents of inorganic sulfide and organic sulfide increased. Fifty flavor compounds were detected, including nineteen esters, twelve alcohols, seven ketones, six aldehydes, three alkenes, one furan, one pyridine, and one acid. There were no significant differences in the type or content of flavor compounds. However, PLS-DA showed differences among the samples. Eighteen volatile organic compounds with variable importance in projection values > 1 were obtained. There were sensory differences among the four samples. Compared with the sample fermented with only *S. cerevisiae*, the samples co-fermented with *Lactobacillus* or with *P. pastoris* had an obvious bitter taste and mellow taste, respectively. The sample fermented by all three strains had a prominent fruity flavor. Except for the sample fermented with only *S. cerevisiae,* the jujube flavor was weakened to varying degrees in all samples. Co-fermentation could be a valuable method to improve the flavor quality of distilled jujube liquor. This study revealed the effects of different mixed fermentation modes on the sensory flavor of distilled jujube liquor and provided a theoretical basis for the establishment of special mixed fermentation agents for distilled jujube liquor in the future.

## 1. Introduction

Jujube (*Ziziphus jujuba* Mill) is a plant from the family Rhamnaceae that originated in China [1]. Jujube has been cultivated for more than 4000 years and is grown in Xinjiang, Shandong, Hebei, Gansu, and Ningxia [2,3]. Jujube fruit contains a variety of vitamins, especially vitamin C and polysaccharides [4,5], and is considered a traditional medicinal with nutritional value [6,7]. The fruit usually matures in the autumn and winter. Fresh jujube can be stored for 3–4 days at room temperature, but the storage period can be extended to several weeks by refrigeration [8]. In addition to refrigeration, drying increases the storage period and enhances the flavor characteristics of the fruit [9,10].

A deep-processing method is required to convert dried jujube into alcoholic beverages. Jujube wine with unique flavors can be produced after fermentation. In recent years, researchers have conducted studies on the production technology and quality of jujube wine. For example, Lee et al compared jujube wine brewed from different raw materials and found that drying had a great impact on the quality and flavor of jujube wine [8]. Xia et al. used GC-O-MS to determine the main aroma compounds and odor activity values of jujube brandy. Through the calculation of odor activity values, the authors identified 27 aroma and 16 flavor compounds [11]. Zhao et al., who fermented jujube wine using *Saccharomyces cerevisiae* and *Bacillus licheniformis*, concluded that mixed fermentation effectively improved quality [12]. Xu et al. reported that pulse electric fields significantly enriched the floral and fruity volatile notes of jujube wine [13]. In addition, Cai et al. used HS-SPME-GC-MS to assess the effects of pretreatment and leaching on jujube wine quality. Their findings showed that blended-into-pulp and leached-pectinase treatments significantly improved the quality of jujube wine [14].

The production technology of jujube wine is similar to that of traditional port wine, i.e., it requires raw material pretreatment, leaching, fermentation, and aging [8,14,15]. The production technology results in a jujube wine with a strong jujube flavor but a low ethanol content. In general, a solid-state fermentation technology combined with distillation is used to produce distilled jujube liquor. However, the high levels of pectin in jujube [16] result in an excessive methanol content. Therefore, liquid fermentation coupled with distillation are used to improve the liquid ratio during fermentation. Additionally, pectinase and cellulase are used to reduce pectin in raw materials, thereby reducing their methanol content. Compared with non-distilled jujube wine and distilled jujube liquor produced by solid-state fermentation technology, there is less research on liquid fermentation technology and the special fermentation agents of jujube liquor.

Brandy and Chinese Baijiu are traditional distilled liquors [17,18] that contain water, alcohols, esters, acids, aldehydes, ketones, and aromatic compounds [19,20]. These alcoholic beverages have unique flavor characteristics. Brandy has a strong fruit aroma, while Baijiu has a richer taste and mixed aroma. In this study, to obtain both the fruity flavor of brandy and the rich taste of Baijiu, probiotics [21,22] from Baijiu fermentation were used in the production of a new type of distilled jujube liquor. Sensory evaluation combined with gas chromatography-ion mobility spectrometry (GC-IMS) were used to evaluate the effects of mixed fermentation with *S. cerevisiae* (SC), *S. cerevisiae* and *Lactobacillus* (SC.L), *S. cerevisiae,* and *Pichia pastoris* (SC.PP), *S. cerevisiae*, *P. pastoris*, and *Lactobacillus* (SC.PP. L) on the quality of distilled jujube liquor.

## 2. Materials and Methods

### 2.1. Materials

Jujube (“Changhong jujube” variety) was produced in Zaozhuang, China. *P. pastoris* and *Lactobacillus* were obtained from the distillery, and *S. cerevisiae* was provided by Angel Yeast Co., Ltd. (Yichang, China). Sucrose was obtained from a local market, and the MRS and YPD media were prepared in our laboratory (Zaozhuang, China). Cellulase and pectinase were acquired from Henan Wan Bang Industrial Co., Ltd. (Zhengzhou, China).

### 2.2. Strain and Inoculum Preparation

MRS and YPD media were used to culture *Lactobacillus* (37 °C, 200 r.min^−1^, 48 h) and *P. pastoris* (30 °C, 180 rpm^−1^, 48 h), respectively.

### 2.3. Sample Pretreatment and Fermentation

First, dried jujube was steamed for 1 h. After placing the sample in boiling water and adjusting the liquid ratio to 7:1, the sample was mixed for 10 min. Second, 1% cellulase and 1% pectinase were added, and the homogenate was kept warm for 1 h. Third, the homogenate was filtered, and seeds and peels were removed. The homogenate was transferred into fermentation tanks and numbered 1 through 4. Fourth, the temperature of the fermentation tanks was set to 30 °C, and the sugar content was adjusted to 20°BX. Next, 0.3 g/L *S. cerevisiae* was added to all four tanks; *Lactobacillus* collected after centrifugation (from 1 L medium) was added to tank 2; *P. pastoris* collected after centrifugation (from 1 L medium) was added to tank 3; and *Lactobacillus* and *P. pastoris* collected after centrifugation (from 1 L medium) were added to tank 4. The homogenate was fermented at 25 °C for 32 d. Finally, the fermentation liquid was distilled, the head and tail were removed, the heart was kept, and it was blended to 42% vol.

### 2.4. Physicochemical Analysis

To determine the ethanol in the fermentation liquid, the Chinese national standard GB 5009.225-2016 was followed. The phenol-sulfuric acid method was used to assess residual sugar in the fermentation liquid. Total acid was assessed by titration. The chromaticity of the fermentation liquid was assessed by measuring its absorbance at 425 nm. Using distilled water as the control, the transmittance of the fermentation liquid was measured at 680 nm.

### 2.5. E-Nose Analysis

The E-nose (PEN 3, Airsense Analytics Co. Ltd., Schwerin, Germany) has 10 different metal oxide sensors, and each sensor (receptor) is sensitive to each corresponding compound [23].

A measure of 1 mL of jujube liquor diluted to 10% vol was transferred into the 30 mL test bottle, and a part of the samples was tested immediately, while another part of the samples was tested reversely after standing for 5 min at room temperature.

The E-nose detection was performed after 30 min of equilibration. The conditions of the E-nose were set as follows: sensor cleaning before the test, 100 s; sensor cleaning before each detection, 100 s; E-nose detection time, 100 s; wait time, 5 s; and gas flow rate, 100 mL/min.

The descriptions for the E-nose sensor performance are presented in Table 1.

### 2.6. HS-GC-IMS Analysis

Volatile organic compounds (VOCs) in jujube wine were identified as reported by Liu et al. [24] with some modifications. GC-IMS (FlavourSpec^®^, Gesellschaft für Analytische Sensorsysteme mbH, Dortmund, Germany) equipped with an MXT-WAX column (15 m × 0.53 mm, 0.1 μm, RESTEK, Bellefonte, PA, America) was used. A 1 g sample was placed into a 20 mL headspace vial and incubated at 60 °C for 10 min. A measure of 500 µL of headspace sample was placed into the injector at 85 °C. Nitrogen (99.999% pure) was used as the carrier gas. The gradient profile was the following: 0–2 min, 2 mL/min; 2–10 min, 2–10 mL/min; 10–20 min, 10–100 mL/min; and 20–30 min, 100 mL/min. The total running time was 30 min.

The IMS detector was set to 45 °C. The carrier gas was nitrogen (99.999% pure) at 150 mL/min. GC-IMS used NIST library and IMS library for qualitative analysis.

### 2.7. Sensory Evaluation

Sensory analysis was conducted with 10 trained evaluators (5 men and 5 women, between the ages of 23 and 30). After reviewing the relevant literature [8,12], the sensory descriptors were “sweet”, “fruit”, “jujube aroma”, “alcohol”, “sour”, “bitter”, “green”, and “overall preference”. The intensity was expressed on a 10 point scale (0 = imperceptible; 10 = extremely intense). Each evaluator assessed each sample three times and took the average value as the evaluation result.

### 2.8. Statistical Analysis

TBtools (version 1.1047, South China Agricultural University, Guangzhou, Guangdong, China) was used for the heat maps [25] and SIMCA (version 13.0, Umeå, Sweden) for the partial least squares discrimination analysis (PLS-DA) and Biplot. For one-way analysis of variance, SPSS (version 23.0, Chicago, IL, USA) was used. GC-IMS data were analyzed using VOCal and Reporter.

All samples were tested three times. The results are expressed as mean ± standard deviation.

## 3. Results and Discussion

### 3.1. Physicochemical Analysis

The physicochemical properties of the fermentation liquid are shown in Table 2. There were differences in ethanol, residual sugar, total acid, chromaticity, and transmittance.

The ethanol content of the jujube liquor fermentation broth in SC, SC.L, SC.PP, and SC.L.PP was 11.4%, 10.6%, 9.4%, and 11.2%, respectively. The ethanol content of SC.PP.L was slightly lower than SC, but the difference was not significant, indicating that *Lactobacillus* and *P. pastoris* had a certain impact on the ethanol production capacity of *S. cerevisiae*. The residual sugar content in SC and SC.PP.L was 0.26 and 0.24 g/L, respectively, with no significant differences. The highest content of residual sugar was in SC.PP (0.30 g/L), which had a negative correlation with the ethanol content. In other words, the content of residual sugar with a high ethanol content was relatively low because the sugar was converted into ethanol as a substrate. Wang et al. found that the residual sugar content in the mixed fermentation state was significantly higher than that in the pure *S. cerevisiae* fermentation, and the ethanol content in the pure *S. cerevisiae* fermentation was the highest, which was similar to the results of this study, indicating that the mixed fermentation of *S. cerevisiae* and *non-S. cerevisiae* would lead to the reduction of ethanol production [26]. The highest total acid content was in SC.L (5.48 g/L), followed by SC (5.05 g/L), SC.L.PP (4.88 g/L), and SC.PP (3.93 g/L). *Lactobacillus* converts sugars into lactic acid, so the total acid content in SC.L was the highest. *S. cerevisiae* produces acids during metabolism, while *P. pastoris* inhibits acid production [21,27]. Therefore, total acid was lowest in SC.PP. SC and SC.PP had the lowest and highest chromaticity, respectively. In contrast, transmittance was the highest and lowest in SC and SC.PP, respectively. Therefore, the color and light transmittance of jujube liquor fermentation liquor were affected by the different strains. At the same time, the chromaticity and transmittance of the sample seem to have some relationship with the residual sugar content. When the residual sugar content is low, the light transmittance is higher and the chromaticity is lower.

The author speculates that *P. pastoris* has poor ethanol production capacity, and it will compete with *S. cerevisiae* for carbon source, thus significantly reducing the ethanol concentration. In addition, some studies have shown that acid can reduce the activity of *S. cerevisiae* and even kill it [28], while the interaction between *P. pastoris* and *S. cerevisiae* can reduce the acid content, thus increasing the ethanol yield.

### 3.2. Volatile Compounds in Jujube Liquor Identified via E-Nose

As shown in Figure 1, Figure 1a–d is the E-nose radar chart of four mixed fermentation combinations. The 0 min line is the result of the test immediately after putting jujube liquor into the test bottle. The strength of sensors 6, 7, and 8 in SC is higher, indicating that the content of methyl, inorganic sulfide, alcohols and aldehydes, and ketones in SC is higher. In general, the content of nitrogen oxides, methyl compounds, and inorganic sulfides in jujube liquor is high.

The 5 min line indicates the result of the test of jujube liquor after five minutes at room temperature, which is used to simulate the changes of substances in the process of jujube liquor decanting. Compared with 0 min, the strength of sensors 6 and 8 for jujube wine placed for 5 min decreased, and the strength of sensors 7 and 9 increased significantly. It is speculated that the reason may be that methyl, alcohol and aldehyde, and ketone substances are more volatile, and the loss is greater during the placing process. However, inorganic sulfide and organic sulfide volatilize relatively slowly, so they are detected in large quantities after being placed.

Similar to port wine, the sulfide in jujube liquor is mainly brought into the fermentation system when adding sulfur-containing fungicides. These compounds have a significant impact on port wine aroma attributes and quality [29]. When the sulfide concentration (such as H_2_S) exceeds the safe range, it will impart the aromas of rotten egg and sewage [30]. Similarly, these substances have a negative impact on the quality of jujube liquor, and the content of sulfide can be reduced by decanting.

### 3.3. Analysis of Volatile Components by GC-IMS

#### 3.3.1. GC-IMS Two-Dimensional Spectra

Figure 2a shows the top view of the three-dimensional topographic plot; the abscissa in the figure represents the drift time of the material, and the ordinate represents the retention time. The abscissa 1.0 is the reactive ion peak (RIP), the red line close to it is the ethanol peak, and each point on the right represents a volatile compound. As there are several points in Figure 2a, it is difficult to make a comparison. Therefore, SC was selected as the reference, and as other samples were subtracted from the reference, Figure 2b could be obtained. The red and blue points in the figure indicate that the content of a compound is higher and lower, respectively, than in the reference. The white area indicates that the content of a compound is similar to that in the reference. In the figure, differences among the samples can be observed. The blue spot in the area marked by the green line indicates that the content of the substance is lower than SC, and the red spot in the area marked by the red line indicates that the content of the substance is higher than SC.

#### 3.3.2. Qualitative Analysis of Volatile Components

Appendix A lists the VOCs identified by GC-IMS. The NIST library and IMS library built into GC-IMS were used in the qualitative analysis. Forty-one VOCs (eight monomers and dimers) were detected, including esters (fourteen), alcohols (nine), ketones (seven), aldehydes (six), alkenes (two), a furan (one), a pyridine (one), and an acid (one). Esters accounted for 34.1%, most of which were ethyl esters. The esters were associated with banana, pineapple, and rose aromas in the liquor [19].

Figure 3a–e shows the VOCs. In SC, the most prevalent compounds were butanol, 2-pentanone, acetone, 3-methylbutal, methanol, 1-propanol, isoamyl ace state (D&M), isobutyl acetate (D&M), 2-methyl-1-propanol-D, butyl ace state, and propanal-D. In SC.L, the most predominant compounds were 1-hexanol, 2-methyl-1-propanol-M, 3-methyl-1-butanol-M, ethyl isobutyrate (D&M), ethyl hexanoate, and ethyl lactate. The content of these compounds was relatively high in SC.PP. Alcohols provide a sweet taste [31,32] in liquors, but an excessive alcohol content contributes to a bitter taste [33]. Ethyl formate (D&M), acetaldehyde, terpinolene, and 2-methylpropanol were high in SC.PP.L. Ketones and esters might be related to the addition of *P. pastoris*. In general, SC.PP and SC.PP.L contained more ketones and esters, which is related to the interaction between *P. pastoris* and *S. cerevisiae* during fermentation [21].

### 3.4. PLS-DA Analysis

PLS-DA is a common data analysis method [34]. PLS-DA focuses on the differences between different categories of samples and operates by splitting the hyperspace of the variables. The discrimination rule is based on a comparison of the predicted response values from Y with a fixed scalar threshold (usually 0.5) [35].

Figure 4a shows that different samples could be distinguished. There were obvious differences between SC and SC.PP.L. Figure 4b shows the relationship between samples and VOCs. If the sample point is close to the point of the compound, it indicates that the correlation between the compound and the sample is high, and the content of the compound is higher in the samples. In SC, 3-methylbutal, butanal, 2-butanone, acetone, and 2-pentanone were predominant. 3-methylbutal contributes sweet, fruity, and banana flavors in liquor. Butanol is usually a compound with a stimulating taste, but it will present a fruity flavor when it exists at a lower concentration [36]. Compounds 2-butanone and 2-pentanone have a sweet taste similar to acetone. Ethyl isobutyrate (D&M), acetaldehyde, and 3-methyl-1-butanol-M had a high correlation with SC.L. Ethyl isobutyrate contributes sweet aromas in Baijiu, 3-methyl-1-butanol usually contributes malty, and acetaldehyde at lower concentrations contributes fruity, coffee, wine, and green aromas [37,38]. The substances with a high correlation in SC.PP were 1-butanol (D&M), propyl acetate, 3-hydroxy-2-butanone. Compounds 1-butanol and propyl acetate contribute banana and fruit flavors, while 3-hydroxy-2-butanone has a butter flavor [36,37]. Ethyl formate (D&M), 1-Hexanol, and terpinolene were predominant in SC.PP.L. Both terpinolene and 1-hexanol contribute floral fragrance [39].

In addition, the variable importance in projection (VIP) values of the VOCs were calculated by PLS-DA (Figure 4c). There were 21 compounds with VIP > 1, including 1-hexanol, 3-hydroxy-2-butanone, 3-methyl-1-butanol (D&M), 2-heptanone, ethyl hexanoate, 1-butanol (D&M), 2,3-pentanedione, ethyl 3-methylbutanoate, isobutyl acetate (D&M), 2-pentanone, propyl acetate, ethyl propanoate (D&M), 3-methylbutanal, ethyl acetate, propanal-M, methanol, and 2,6-dimethylpyridine (Table 3). These VOCs may be the main reason for the differences among samples.

### 3.5. Sensory Evaluation

As shown in Figure 5, mixed fermentation has a significant impact on the sense of jujube liquor. “Overall preference” decreased in SC.L and SC.PP, while “bitter” and “green” increased. A strong “green” is associated with a strong pine branch flavor. “Jujube aroma” in the samples weakened after mixed fermentation. In SC, “jujube aroma” and “sweet” were predominant. Jujube aroma originates from raw materials and enters jujube liquor during distillation. Ethanol contributes to sweetness [40]. However, liquors with the same ethanol content often differ in sweetness; therefore, ethanol is not the determinant of sweetness in liquors. Some esters impart strong sweetness [17]. Reports have shown that an appropriate content of higher alcohols increases the sweetness of Chinese Baijiu [31,32]. Research shows that some aroma compounds can enhance the human body’s sense of sweetness at the sensory level, thereby enhancing the flavor of drinks and reducing sugar intake [41]. The general outlines of SC.L and SC.PP were similar, with differences in “sweet” and “sour”. In general, the SC.PP.L group made the sensory indexes of jujube wine more coordinated, thus obtaining the highest score in Overall Performance. Similarly, Zhang et al. found that the combined fermentation of multiple microorganisms played a positive role in improving the quality of fruit wine when studying the fermentation of mulberry wine with different ratios by *Lactobacillus plantarum* and *Oenococcus oeni* [42].

## 4. Conclusions

Jujube liquor was produced by mixed fermentation with *P. pastoris*, *Lactobacillus,* and *S. cerevisiae*, which were screened from the fermentation broth of Chinese Baijiu. The analysis of routine physical and chemical indexes showed that the combination of different strains had obvious or even significant differences. Ethanol levels decreased, *Lactobacillus* increased the total acid content, and *P. pastoris* reduced the total acid content. An E-nose was used to detect the changes of volatile components in the process of decanting, and it was found that the contents of methyl, alcohol, and aldehyde ketones were significantly reduced, while the contents of inorganic sulfide and organic sulfide were increased. The effects of mixed fermentation on the quality of distilled jujube wine were compared by GC-IMS and sensory evaluation. The samples were clustered by PLS-DA. Fifty compounds were detected, including nineteen esters, twelve alcohols, seven ketones, six aldehydes, three alkenes, one furan, one pyridine, and one acid. Eighteen VOCs with VIP > 1 were obtained. The sensory evaluation results showed that there were sensory differences among the four samples. Compared with SC, SC.L, and SC.PP had a bitter taste and mellow taste, respectively, while the fruit aroma of SC.PP.L was prominent. In addition, the jujube flavor of the samples, with the exception of SC, was weakened to varying degrees. Above all, the results show that different microbial combinations have an impact on the quality of distilled jujube liquor. These combination strategies can be applied to improve the sensory characteristics of distilled jujube liquor, but the types and mixing proportion of microorganisms need to be further optimized until a special starter suitable for distilled jujube liquor can be established.

## Figures and Tables

**Figure 1 foods-12-00795-f001:**
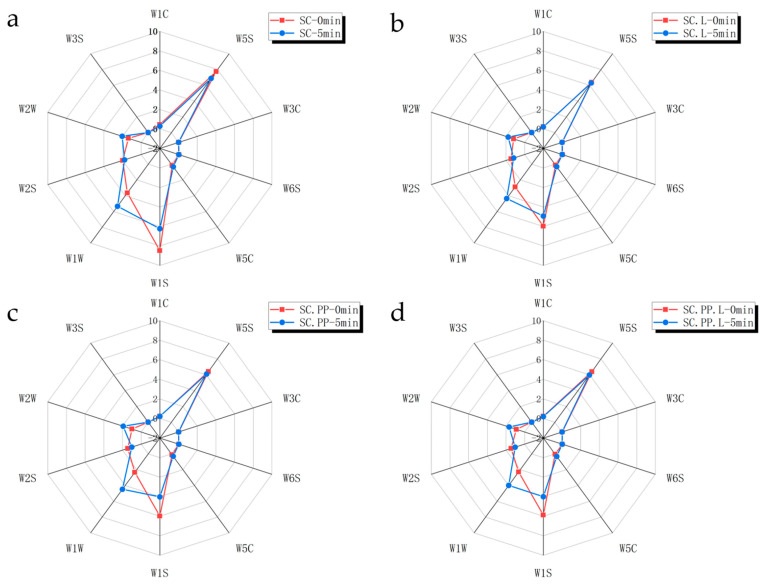
E-nose radar chart of different groups of jujube liquor. (**a**) Comparison chart of SC; (**b**) Comparison chart of SC.L; (**c**) Comparison chart of SC.PP; (**d**) Comparison chart of SC.PP.L; (*S. cerevisiae*-‘SC’, *S. cerevisiae* and *Lactobacillus*-‘SC.L’, *S. cerevisiae*, and *P. pastoris*-‘SC.PP’, *S. cerevisiae*, *P. pastoris*, and *Lactobacillus*-‘SC.PP. L’; 0min-test immediately after putting the sample into the test bottle, 5min-test after standing the sample at room temperature for 5 min).

**Figure 2 foods-12-00795-f002:**
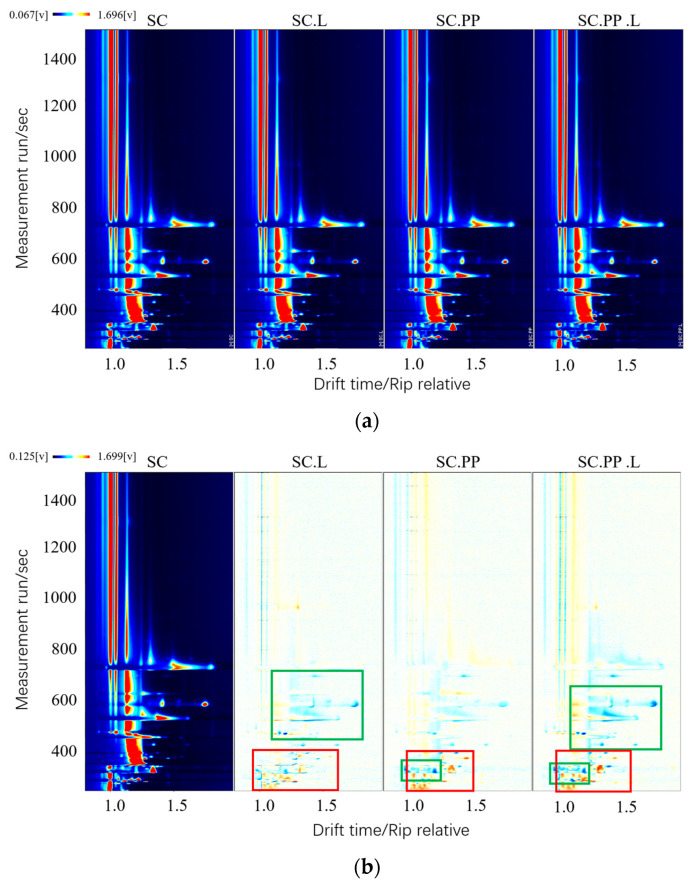
GC-IMS Two-Dimensional Spectra. (**a**) Intuitive comparison Spectra; (**b**) difference comparison Spectra. (*S. cerevisiae*-‘SC’, *S. cerevisiae* and *Lactobacillus*-‘SC.L’, *S. cerevisiae*, and *P. pastoris*-‘SC.PP’, *S. cerevisiae*, *P. pastoris*, and *Lactobacillus*-‘SC.PP. L’).

**Figure 3 foods-12-00795-f003:**
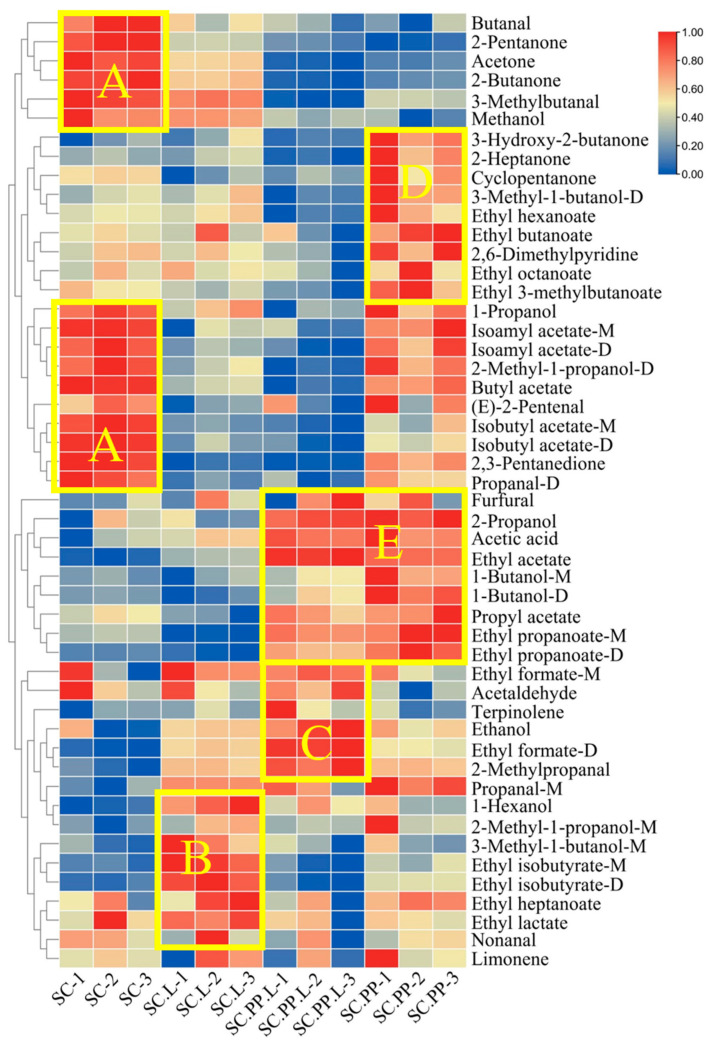
Heatmap analysis of volatiles taken from GC-IMS of the Jujube liquor. (**A**) Compounds with higher content in SC; (**B**) Compounds with higher content in SC.L; (**C**) Compounds with higher content in SC.PP.L; (**D**) Compounds with higher content in SC.PP; (**E**) Compounds with higher content in SC.PP.L and SC.PP; (*S. cerevisiae*-‘SC’, *S. cerevisiae* and *Lactobacillus*-‘SC.L’, *S. cerevisiae*, and *P. pastoris*-‘SC.PP’, *S. cerevisiae*, *P. pastoris*, and *Lactobacillus*-‘SC.PP. L’).

**Figure 4 foods-12-00795-f004:**
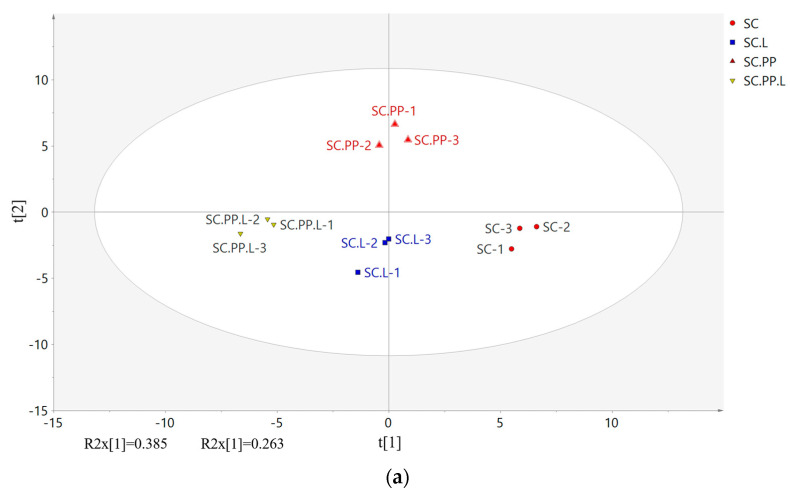
(**a**) Partial least squares Discriminant Analysis (PLS-DA) plot of different groups of jujube liquor; (**b**) loading plot of different groups of jujube liquor; (**c**) VIP plot of different groups of jujube liquor. (*S. cerevisiae*-‘SC’, *S. cerevisiae* and *Lactobacillus*-‘SC.L’, *S. cerevisiae*, and *P. pastoris*-‘SC.PP’, *S. cerevisiae*, *P. pastoris*, and *Lactobacillus*-‘SC.PP. L’).

**Figure 5 foods-12-00795-f005:**
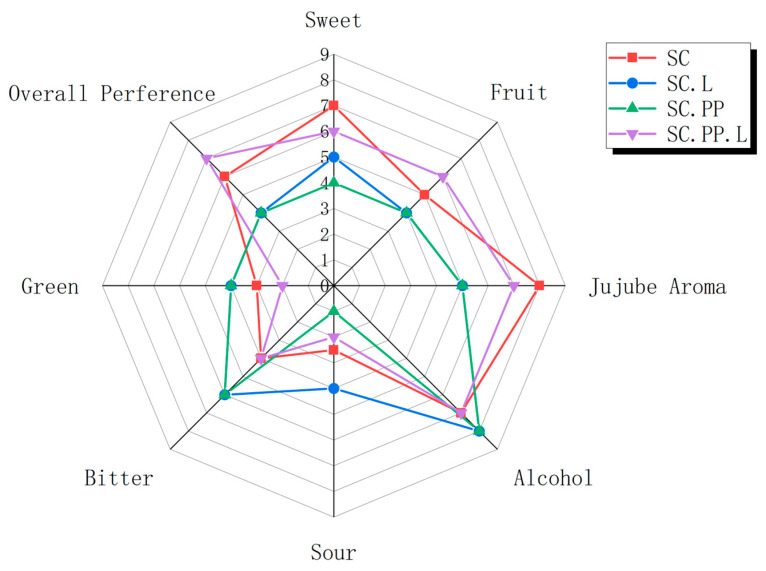
Sensory evaluation radar chart of different groups of jujube liquor. (*S. cerevisiae*-‘SC’, *S. cerevisiae* and *Lactobacillus*-‘SC.L’, *S. cerevisiae*, and *P. pastoris*-‘SC.PP’, *S. cerevisiae*, *P. pastoris*, and *Lactobacillus*-‘SC.PP. L’).

**Table 1 foods-12-00795-t001:** Performance characteristics of E-Nose sensor.

Number	Receptor	General Description
1	W1C	Aromatic compounds
2	W5S	Very sensitive to nitrogen oxides
3	W3C	Ammonia, used as sensor for aromatic compounds
4	W6S	Mainly hydrogen, selectively (breath gases)
5	W5C	Alkenes, aromatic compounds, less polar compounds
6	W1S	Sensitive to methane broad range
7	W1W	Reacts on sulphur compounds
8	W2S	Detects alcohols, partially aromatic compounds
9	W2W	Aromatics compounds, sulphur organic compounds
10	W3S	Reacts on high concentrations

**Table 2 foods-12-00795-t002:** Physicochemical properties of fermentation liquid.

Samples	Ethanol/%	Residual Sugar/(g/L)	Total Acid/(g/L)	Chromaticity	Transmittance/%
**SC**	11.4 ± 0.11 ^a^	0.26 ± 0.00 ^ab^	5.05 ± 0.25 ^b^	0.21 ± 0.00 ^c^	56.89 ± 0.31 ^a^
**SC.L**	10.6 ± 0.10 ^b^	0.29 ± 0.00 ^b^	5.48 ± 0.23 ^a^	0.28 ± 0.00 ^b^	45.74 ± 0.92 ^c^
**SC.PP**	9.4 ± 0.15 ^c^	0.30 ± 0.00 ^b^	3.93 ± 0.35 ^c^	0.32 ± 0.00 ^a^	42.76 ± 0.96 ^d^
**SC.PP.L**	11.2 ± 0.11 ^a^	0.24 ± 0.00 ^a^	4.88 ± 0.31 ^b^	0.23 ± 0.00 ^c^	53.23 ± 0.28 ^b^

Note: Different letters in the same column represent statistically significant differences (*p* < 0.05).

**Table 3 foods-12-00795-t003:** Volatile compounds with VIP > 1.

Count	Compound	CAS	Aroma Descriptors ^a^
1	1-Hexanol	111-27-3	Banana, Flower, Grass, Herb
2	3-Hydroxy-2-butanone	513-86-0	Butter, Creamy, Green Pepper
3	3-Methyl-1-butanol-D	123-51-3	Burnt, Cocoa, Floral, Malt
4	3-Methyl-1-butanol-M	123-51-3	Burnt, Cocoa, Floral, Malt
5	2-Heptanone	110-43-0	Blue Cheese, Fruit, Green, Nut, Spice
6	Ethyl hexanoate	123-66-0	Apple Peel, Brandy, Fruit Gum, Pineapple
7	1-Butanol-D	71-36-3	Fruit
8	1-Butanol-M	71-36-3	Fruit
9	2,3-Pentanedione	600-14-6	Butter, Cream
10	Ethyl 3-methylbutanoate	108-64-5	Apple, Fruit, Pineapple, Sour
11	Isobutyl acetate-D	110-19-0	Apple, Banana, Floral, Herb
12	Isobutyl acetate-M	110-19-0	Apple, Banana, Floral, Herb
13	2-Pentanone	107-87-9	Fruit, Pungent
14	Propyl acetate	109-60-4	Celery, Floral, Pear, Red Fruit
15	Ethyl propanoate-D	105-37-3	Apple, Pineapple, Rum, Strawberry
16	Ethyl propanoate-M	105-37-3	Apple, Pineapple, Rum, Strawberry
17	3-Methylbutanal	590-86-3	Fruity, dry green, chocolate, nutty, cocoa
18	Ethyl acetate	141-78-6	Aromatic, Brandy, Contact Glue, Grape
19	Propanal-M	123-38-6	Floral, Pungent, Solvent
20	Methanol	67-56-1	Alcoholic
21	2,6-Dimethylpyridine	108-48-5	Nutty, Coffee, Cocoa, Musty, Bready

Note: ^a^ From Flavornet database (https://www.femaflavor.org; http://www.flavornet.org; http://www.thegoodscentscompany.com; accessed on 3 February 2023).

## Data Availability

Data is contained within the article or Supplementary Material.

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
