# Peer review of "Analysis of the Effect of Mixed Fermentation on the Quality of Distilled Jujube Liquor by Gas Chromatography-Ion Mobility Spectrometry and Flavor Sensory Description"

_foods, 2023, doi:10.3390/foods12040795_

Round 1

Reviewer 1 Report

Manuscript foods-2173551 concerns the production  of jujube wine using different yeast species. The manuscript drives a novel hypothesis and is an original contribution. The preparation of novel alcoholic beverages is of great importance for consumers or the food sector. The manuscript falls within the aims and scope of Foods journal. In general, it has been well designed providing volatile compounds and sensory analysis data of the  fermented product identified using the most recent GC-IMS technique. Numerical data have been well treated with chemometrics

There are however important spelling formats that must be corrected to improve the description of the work carried out. I have indicated within the attached pdf the p[roblems raised throygh the review of this work.

Based on my overall comments, I suggest a revision prior to further consideration.

Reviewer 2 Report

Comments to the authors:

-          Keywords: Please use words that are different from the words in the title.

-          The Abstract does not follow the general structure recommended. I suggest rewriting the abstract by structuring it as: Background, Objectives (state briefly the novelty of the study), a brief statement for Methods, major results, a brief conclusion, and the significance of your research.

-          The necessity of doing research is not mentioned. The gap in research should be described clearly.

-          Figure and table captions should be rewritten because they do not explain the subject completely. The captions must be independent of the text.

-          The resolution of the Figure 3(b and c) is not sufficient.

-          Figure 4 is not cited in the text.

-          It should be revised the discussion part of the results majorly with more explanation and

supporting references.

-          Please add some suggestions for future trends at the end of "Conclusion".

-          The English language needs to be revised and grammatical errors should be removed.

Reviewer 3 Report

Manuscript No.: foods-2173551

Title: Analysis of the effect of mixed fermentation on the quality of distilled jujube liquor by GC-IMS and flavor sensory description.

I have completed the review of the paper that you have sent to me to review it. I have reviewed your manuscript, and I am expressing my positive feedback. Your study is of importance for the readers of the Journal. However, before recommending your manuscript to be accepted for publication in this fine journal, I am requesting the minor revisions according to the following comments:

1.      What is the future scope of the present data?

2.      Novelty of work is missing, the authors must shows this.

3.      What is the reason for selected the mixed fermentation for this study?

4.      The introduction is very long, it takes up about a quarter of the paper, I suggest to the authors to minimize it as much as possible. In introduction some suitable recent references may be cited such as DOI:10.3390/foods10112571,

5.      The manuscript is excessively long and detailed. Much of the discussion could be made more succinct and focused more on the key new results.

6.       Table 2 is very big, I suggest to the authors to keep only the necessary results or moved it to supplementary information.

7.       The quality of the figures is very poor, that's why I ask the authors to improve the quality of all the figures.

8.      Conclusion part should be modified to explain the important findings in this article.

9.      Expand all abbreviations when it first appears in the manuscript.

10.  Grammatical errors should be checked carefully.

11.  Check the reference format.

Once you address all of the above-mentioned comments, I will gladly review your manuscript again.- Minor revision

Round 2

Reviewer 2 Report

-          The revised manuscript has considerably improved. I accept the corrections and explanations that the authors have made to the manuscript. In my opinion, the manuscript could be published.